# The Effects of Calculated Remnant-Like Particle Cholesterol on Incident Cardiovascular Disease: Insights from a General Chinese Population

**DOI:** 10.3390/jcm10153388

**Published:** 2021-07-30

**Authors:** Yanli Chen, Guangxiao Li, Xiaofan Guo, Nanxiang Ouyang, Zhao Li, Ning Ye, Shasha Yu, Hongmei Yang, Yingxian Sun

**Affiliations:** 1Department of Cardiology, the First Hospital of China Medical University, Shenyang 110001, China; ylchen@cmu.edu.cn (Y.C.); xfguo@cmu.edu.cn (X.G.); 20082174@cmu.edu.cn (N.O.); lizhao@cmu.edu.cn (Z.L.); 2016110112@cmu.edu.cn (N.Y.); ssyu@cmu.edu.cn (S.Y.); hmyang@cmu.edu.cn (H.Y.); 2Department of Medical Record Management Center, the First Affiliated Hospital of China Medical University, Shenyang 110001, China; ligx1@cmu1h.com

**Keywords:** remnant cholesterol, diabetes mellitus, cardiovascular disease, dyslipidemia

## Abstract

Background: Growing evidence suggests that remnant cholesterol (RC) contributes to residual atherosclerotic cardiovascular disease (ASCVD) risk. However, the cutoff points to treat RC for reducing ASCVD are still unknown. This study aimed to investigate the relationships between RC and combined cardiovascular diseases (CVDs) in a general China cohort, with 11,956 subjects aged ≥ 35 years. Methods: Baseline RC was estimated with the Friedewald formula for 8782 subjects. The outcome was the incidence of combined CVD, including fatal and nonfatal stroke and coronary heart disease (CHD). The Cox proportional hazards model was used to calculate hazard ratios (HRs) with 95% confidence intervals. The restricted cubic spline (RCS) model was used to evaluate the dose–response relationship between continuous RC and the natural log of HRs. Results: After a median follow-up of 4.66 years, 431 CVD events occurred. In the Cox proportional models, participants with a high level of categorial RC had a significantly higher risk for combined CVD (HR: 1.37; 95% CI: 1.07–1.74) and CHD (HR: 1.63; 95% CI: 1.06–2.53), compared to those with a medium level of RC. In the stratification analyses, a high level of RC significantly increased combined CVD risk for subgroups females, age < 65 years, noncurrent smokers, noncurrent drinkers, normal weight, renal dysfunction, and no hyperuricemia. The same trends were found for CHD among subgroups males, age < 65 years, overweight, renal dysfunction, and no hyperuricemia; stroke among subgroup females. In RCS models, a significant linear association between RC and combined CVD and a nonlinear association between RC and CHD resulted. The risk of outcomes was relatively flat until 0.84 mmol/L of RC and increased rapidly afterwards, with an HR of 1.308 (1.102 to 1.553) for combined CVD and 1.411 (1.061 to 1.876) for CHD. Stratified analyses showed a significant nonlinear association between RC and CVD outcomes in the subgroup aged < 65 years or the diabetes subgroup. Conclusions: In this large-scale and long-term follow-up cohort study, participants with higher RC levels had a significantly worse prognosis, especially for the subgroup aged 35–65 years or the diabetes mellitus subgroup.

## 1. Introduction

Atherosclerotic cardiovascular disease (ASCVD) is a leading cause of morbidity and mortality worldwide and remains a major public health challenge [1]. Lipid abnormalities play a central role in the pathogenesis of ASCVD [2]. Lowering plasma levels of low-density lipoprotein (LDL) cholesterol (LDL-C) has been reported as an important modality to prevent ASCVD for decades [3,4]. However, substantial residual risk remains despite achieving an LDL-C level as a mean of 30–40 mg/dL with statins, ezetimibe, and/or proprotein convertase subtilisin/kexin type 9 (PCSK9) inhibitors [5]. Due to the negative results of high-density lipoprotein (HDL)-raising drugs (niacin, cholesteryl ester transfer protein inhibitors), regulating other lipid components to reduce the residual risk of ASCVD has become the new focus for lipid intervention [6]. The Reduction of Cardiovascular Event Icosapent Ethyl Intervention Trial (REDUCE-IT) found that a reduction in triglycerides by 20% led to a 25% reduction in atherosclerotic cardiovascular events [7]. However, roughly one-half of the risk reduction can be explained by a reduction in remnant cholesterol (RC) [8].

In recent years, growing evidence has suggested that RC is a causal risk factor for cardiovascular events and all-cause mortality [9,10,11,12]. RC is the cholesterol content of triglyceride-rich lipoproteins (TRLs), which are formed when TRLs are partly depleted of triglyceride (TG) by lipoprotein lipase and are composed of very-low-density (VLDLs) and intermediate-density lipoproteins (IDLs) in the fasting state and these two lipoproteins together with chylomicron remnants in the nonfasting state [13]. The cholesterol content of remnant lipoproteins is defined as remnant-like particle cholesterol (RLP-C) [14].

However, the starting point and the target to treat RC for reducing cardiovascular diseases (CVD) are still unknown. More studies are needed to investigate the association between RC with CVD events and further determine the individual target of RC for primary and secondary prevention of CVD risk. This study aimed to investigate the relationships between RC and combined CVDs in a large general Chinese population.

## 2. Materials and Methods

### 2.1. Study Design and Population

The Northeast China Rural Cardiovascular Health Study (NCRCHS) is a community-based prospective cohort study conducted in rural areas of Northeast China. The study design has been described previously [15]. From January 2012 to August 2013, a total of 11,956 subjects aged ≥ 35 years were recruited as a baseline visit from three counties (Dawa, Zhangwu, and Liaoyang) in Liaoning province, using a multistage, randomly stratified cluster-sampling scheme. Detailed information was collected for each subject. In 2015 and 2017, all subjects were invited to attend two stages of follow-up. Detailed cardiovascular examination was repeated in 2015, and incident CVD events were collected in 2017–2018. Of the 11,956 subjects, 10,700 participants consented and qualified for our follow-up study. A total of 10,349 participants completed at least one follow-up visit. The study was approved by the Ethics Committee of China Medical University (Shenyang, China). Written informed consent was obtained from each participant, in accordance with the principles of the Declaration of Helsinki.

In the present analyses, we excluded participants with CVD at baseline (*n* = 821), missing a baseline lipid profile (*n* = 73), or using lipid-lowering agents at baseline (*n* = 248). Participants with ineligible values of HDL cholesterol (HDL-C), LDL-C, and TG (>4.5 mmol/L) were also excluded (*n* = 415). Data were therefore available for 8782 participants. The flow chart of the study is shown in Appendix A online.

### 2.2. Study Variables

At baseline, detailed information on demographic characteristics, dietary and lifestyle factors, and medical history was obtained by an interview with a standardized questionnaire. History of coronary heart disease (CHD) was defined by a self-reported history of myocardial infarction (MI) or prior coronary revascularization and electrocardiography evidence of MI at baseline and confirmed by medical records. History of stroke was similarly defined by a self-reported history of stroke at baseline and confirmed by medical records. Current use of antihypertensive medications, hypoglycemic agents, and lipid-lowering agents was self-reported. Medication use over the prior 2 weeks was verified by reviewing the medication containers that participants brought to the visit. Lipid-lowering agents included statins, bile sequestrants, fibrates, niacin, and antihyperlipidemic medications. Family history of hypertension, diabetes mellitus, stroke, and CHD was defined such that more than one first-degree relative was involved.

All study participants underwent a physical examination, including measurements of weight and standing height and systolic and diastolic blood pressure, as previously reported in detail [15]. Body mass index (BMI) was calculated as weight in kilograms divided by the square of height in meters. According to the World Health Organization (WHO) criteria [16], low and normal weight, overweight, and obesity were defined as BMI < 25 kg/m^2^, BMI 25–29.9 kg/m^2^, and BMI ≥ 30 kg/m^2^, respectively. Blood pressure was measured three times with participants seated after at least 5 min rest using a standardized automatic electronic sphygmomanometer (HEM-907; Omron, Tokyo, Japan). Hypertension (HTN) was defined as systolic blood pressure (SBP) ≥ 140 mm Hg, diastolic blood pressure (DBP) ≥ 90 mm Hg, and/or use of antihypertensive medications. Smoking and alcohol use status was defined as “current use” or not.

### 2.3. Other Variates

Fasting blood samples were collected in the morning after at least 12 h fasting for all subjects. Biochemical analyses were performed automatically (Olympus AU 640, Tokyo, Japan) for the measurement of circulating concentrations of fasting blood glucose (FBG), total cholesterol (TC), LDL-C, HDL-C, TG, uric acid (UA), serum creatinine, and other routine blood biochemical indexes. As previously proposed, fasting RC was calculated as TC minus HDL-C minus LDL-C and expressed as mmol/L [17]. Diabetes mellitus (DM) was defined according to the WHO criterion [18]: FBG ≥ 7.0 mmol/L (126 mg/dL), self-reported physician diagnosis, or use of antidiabetic medications. Estimated glomerular filtration rate (eGFR) was calculated using the Modification of Diet in Rural Disease (MDRD) equation as previously proposed [19]. Renal dysfunction was defined as reduced eGFR < 60 mL/min/1.73 m^2^. Hyperuricemia was defined as serum uric acid level > 420 μmol/L (7 mg/dL) in males and >360 μmol/L (>6 mg/dL) in females [20].

### 2.4. Outcome Ascertainment

The outcome of the present study was CVD incidence. Incident CVD was defined as fatal and nonfatal stroke and CHD according to adjudication by a physician panel. The specific incidences of all-cause mortality, stroke, and CHD were also determined. Hospital records or death certificates were also collected. The diagnosis was classified and coded according to the International Classification of Diseases-Tenth Revision (ICD-10). Stroke was defined following the WHO Multinational Monitoring of Trends and Determinants in Cardiovascular Disease criteria [21] as rapidly developing signs of focal or global disturbance of cerebral function lasting more than 24 h (unless interrupted by surgery or death) with no apparent nonvascular cause. The ICD-10 codes for stroke were I60.x–I69.x. Transient ischemic attack and chronic cerebral vascular disease were excluded. CHD was defined as a diagnosis of hospitalized angina pectoris, hospitalized MI, any coronary revascularization, or CHD death [22], and the ICD-10 codes for CHD were I20.x–25.x. All materials were independently evaluated and adjudicated by the endpoint assessment committee. The date of the first participant recruited was January 2012, and the last follow-up date was January 2018. A series of irregular follow-up visits were conducted. Follow-up time was calculated as the interval between the date of randomization and the date of death, the date of the last visit, or the last recorded clinical event of participants still alive, whichever occurred first.

### 2.5. Statistical Analysis

Data are reported as means and standard deviations for normally distributed variables or as medians for non-normally distributed variables. According to the 33.3% and 66.6% percentiles of RC level, all subjects were divided into three groups (Tertile I, Tertile II, and Tertile III). Baseline characteristics were assessed across different groups with the analysis of variance for parametric variables or Kruskal–Wallis test for nonparametric variables. Bonferroni post-hoc analysis was performed to determine the specific demographic categories that had statistically significant differences. Nominal variables were expressed as absolute numbers and proportions and compared by a *χ*^2^ test or Fisher’s exact test when appropriate. The correlations between continuous RC and adjustment variables were assessed by Pearson’s or Spearman’s rank correlation test as applicable.

The cumulative incidence for cardiovascular outcomes in subjects with different levels of calculated RC at baseline was evaluated by Kaplan–Meier curves and compared with the log-rank test. Cox proportional hazards models were used to calculate hazard ratios (HRs) with 95% confidence intervals (CIs) for the associations between RC (considered as both continuous and categorical variables) and CVD events. Proportionality of hazards was assessed for each variable by using Schoenfeld residuals. Three adjustment models were found for Cox analysis. Model 1 was adjusted for age, gender, and ethnicity. Model 2 was further adjusted for current smoking, current drinking, and BMI (normal, overweight, or obesity). Model 3 was further adjusted for serum levels of triglyceride (continuous), eGFR (dichotomous), uric acid (dichotomous), DM, and HTN.

Restricted cubic spline (RCS) regression [23] was applied to flexibly model the nonlinear association between continuous RC and the natural log of HRs (lnHRs) of outcomes. The knots were placed at the 5th, 50th, and 95th percentiles. Variables in Cox Model 3 were adjusted. Stratified analyses were used to explore whether the association of RC with CVD risk varied across age and DM. Departure from linearity of the final cubic spline model was assessed using the Wald test for nonlinearity [24]. The median value of RC was considered as the reference due to the skew distribution of RC. As the associations of RC and outcomes were approximately log-linear below and above their medians (50th percentile), a linear model was used to calculate HRs per standard deviation increase of RC in outcome prediction [25].

Statistical analyses were performed using SPSS (version 23.0; IBM, Chicago, IL, USA), SAS (version 9.3; Institute Inc., Cary, NC, USA), and GraphPad prism (V.8.4.2; San Diego, CA, USA). Two-sided *p* < 0.05 were considered statistically significant. *p*-Values were adjusted for multiple comparisons using the Bonferroni correction.

## 3. Results

### 3.1. Baseline Characteristics

The characteristics of the participants were assessed by the tertiles of RC in Table 1. In the whole cohort of 8782 subjects (46.4% males), the mean age was 53.2 ± 10.4 years. The mean BMI was 24.7 ± 3.6 kg/m^2^; 47.3% had HTN, and 8.4% had DM.

The mean concentration of RC was 0.8 ± 0.4 mmol/L. According to the level of RC, all participants were divided into three groups: Tertile I (RC < 0.65 mmol/L), Tertile II (RC 0.65–1.00 mmol/L), and Tertile III (RC ≥ 1.00 mmol/L). Across tertiles of RC, age, serum levels of TC, TG, uric acid, and proportion of participants with DM increased gradually (all *p* < 0.001), whereas the level of eGFR, proportions of male, participants with HTN, and current smokers significantly declined (all *p* < 0.05). Serum level of HDL-C and the proportion of participants with current use of alcohol were significantly higher (both *p* < 0.001) in Tertile I than those of the other two groups (*p* < 0.001). Post-hoc analysis with Bonferroni correction revealed a significantly higher age for participants in Tertile III, LDL-C level, than those in Tertile II or Tertile I (*p* < 0.001). Significantly increased levels of TC, TG, and serum uric acid and declined levels of eGFR across tertiles of RC were found.

### 3.2. Survival Analyses for Different Levels of RC

After the median follow-up time of 4.66 years, a total of 431 CVD events occurred in the studied population (293 stroke cases, 150 CHD cases, and 71 MI cases), including 148 fatal CVD cases. The Kaplan–Meier curves for each endpoint in participants with different levels of RC are shown in Figure 1. Participants with high level of RC (Tertile III) had a significantly higher cumulative incidences of combined CVD (*p* = 0.0019), CHD (*p* = 0.0101), stroke (*p* = 0.0448), and fatal CVD (*p* = 0.0465) compared to those with a medium level of RC (Tertile II).

Table 2 shows the multivariable-adjusted HRs for the incidence of outcomes by RC concentration. In the categorial analysis of RC, risks for combined CVD (HR: 1.37; 95% CI: 1.07–1.74) and CHD (HR: 1.63; 95% CI: 1.06–2.53) were significantly higher among participants in Tertile III, compared with those in Tertile II after full adjustment. In the continuous analysis of RC, a high level of RC significantly increased 28% risk of combined CVD (HR: 1.28; 95% CI: 1.02–1.62) and 51% risk of fatal CVD (HR: 1.51; 95% CI: 1.05–2.17) after full adjustment. Significantly higher stroke risk was found for participants in Tertile III after adjustment for Model 1 (HR: 1.31; 95% CI: 1.03–1.67) and Model 2 (HR: 1.30; 95% CI: 1.02–1.67), but the significance disappeared for Model 3. The results were similar when the intensity of physical activity was additionally adjusted.

### 3.3. Stratification Analyses

The relationships between RC and outcomes were evaluated among different RC categories (Tertile II as reference) and stratified with sex, age (<65 years vs. ≥65 years), smoking status (current smoker or not), drinking status (current drinker or not), BMI subgroups (normal, overweight, obesity), DM (yes or no), HTN (yes or no), renal dysfunction (eGFR <60 mL/min/1.73 m^2^ vs. eGFR ≥60 mL/min/1.73 m^2^), and hyperuricemia (yes or no). Due to the small number of other nationalities except Han, there were no subgroup comparisons for nationality.

After full adjustment of other factors, subjects with a high level of RC (Tertile III) had a significantly higher incidence of CVD than those with a medium level of RC (Tertile II) in the following subgroups (Figure 2): females (HR: 1.515, 95%CI: 1.043–2.201), subjects aged ˂ 65 years (HR: 1.547, 95%CI: 1.127–2.123), noncurrent smokers (HR: 1.482, 95%CI: 1.064–2.062), noncurrent drinkers (HR: 1.431, 95%CI: 1.079–1.898), subjects with normal BMI (HR: 1.486, 95%CI: 1.046–2.110), renal dysfunction (HR: 1.357, 95%CI: 1.050–1.752), and normal uric acid (HR: 1.439, 95%CI: 1.101–1.880). The same trends were found for CHD among subgroups: aged ≤ 65 years, overweight subgroup, renal dysfunction, and normal uric acid (Appendix A). For participants in Tertile III, males had a significantly higher risk for CHD (HR: 2.170, 95%CI: 1.038–4.537), and females had more probability of stroke (HR: 1.643, 95%CI: 1.010–2.686) compared to those in Tertile II (Appendix A). There were no significant interactions between subgroups and RC on the incidence of the above outcomes (all *p* > 0.05).

Compared with the reference (Tertile II), a low level of RC (Tertile I) also increased the risk for CHD in subgroups aged < 65 years, no current smoking, no current drinking, BMI 25–30 kg/m^2^, renal dysfunction, and normal uric acid (Appendix A). No significant differences were found for fatal CVD among participants in Tertile III and combined CVD, stroke, and fatal CVD among participants in Tertile I. No significant interactions between subgroups and RC were found on the incidence of outcomes (all *p* > 0.05).

### 3.4. Dose–Response Analyses of RC with Cardiovascular Outcomes

Dose–response analyses were implemented with RCS to investigate the optimal level of RC for each outcome (Figure 3). Multivariable-adjusted RCS analyses showed significant overall associations between RC and combined CVD (*p*_overall_ = 0.0324) or CHD (*p*_overall_ = 0.0398) for the whole cohort. A significant linear relationship (*p*_linear_ = 0.0225) between RC and combined CVD and a nonlinear relationship (*p*_nonlinear_ = 0.0221) between RC and CHD was found for all participants. The risk of combined CVD was relatively flat until around 0.84 mmol/L (32.76 mg/dL) of RC and then started to increase rapidly afterwards. For participants with a higher level of RC than 0.84 mmol/L, the HR of combined CVD and CHD per standard deviation was 1.308 (1.102 to 1.553) and 1.411 (1.061 to 1.876), respectively.

Nonlinear associations between RC and lnHRs for outcomes stratified by subgroups (<65 years vs. ≥65 years; DM or not) are shown in Figure 4. Significant nonlinearity association and “J”-shape curves were shown for RC and combined CVD (*p*_nonlinear_ = 0.0059) or CHD (*p*_nonlinear_ = 0.0002) in participants with an age < 65 years (Figure 4A). Meanwhile, a linear association was found between RC and lnHR for fatal CVD (*p*_linear_ = 0.0104). A 50.7% increase risk for combined CVD, a 57.0% increase risk for CHD, and a 2.0% increase risk for fatal CVD were found for participants under 65 years with a serum level of RC > 0.83 mmol/L. The same trend was found for participants with DM and a serum level of RC > 0.95 mmol/L (Figure 4B). Some significant associations were also found among other subgroups (Appendix A).

## 4. Discussion

The current study of a general Chinese population confirmed an independent relationship between calculated RC and the incidence of CVD. The Kaplan–Meier curves explored significantly higher cumulative incidences of combined CVD, CHD, stroke, and fatal CVD for participants with a high level of RC than those with a medium level of RC. Cox models confirmed a 37% increased risk for combined CVD and a 63% increased risk for CHD in the categorial analysis and a 28% increased risk of combined CVD and a 51% increased risk of fatal CVD in the continuous analysis of RC. A significantly higher risk of stroke was also found for participants with an RC level of Tertile III after part adjustment. Further dose–response association analyses and subgroup analyses elucidated significant associations between continuous RC and the incidence of combined CVD and CHD not only for the whole population but also for the subgroup aged < 65 years or the DM subgroup.

RC refers to the cholesterol content of TRLs, which is composed of chylomicron remnants, VLDL, and IDL. A recent study showed a close relationship between RC and coronary atheroma progression independent of conventional lipid parameters [11]. A growing number of population studies, epidemiological, and genetic evidence suggests that high concentrations of RC are closely associated with a high risk of ischemic heart disease [9,26,27], MI [28,29], and all-cause mortality [10,30]. A cardiovascular benefit of statin therapy, independently of LDL-C reduction, was suggested in reducing TRL-C levels among those with high TRL-C levels [31].

The onset age of CVD is getting younger and younger worldwide [2,32]. Dyslipidemia is a prominently traditional risk factor in the development of early-onset CVD [33,34]. Early identification and modification of atherogenic dyslipidemia can improve primary and secondary prevention of CVD outcomes. The ideal level of lipids is very important in reducing the risk of early-onset cardiovascular-related death [35,36]. Lifestyle modification at the age of 20 years could change the odds of atherosclerosis 30 years later [37]. The currently employed CVD risk assessment tools are heavily age weighted and have been shown to underestimate CVD risk in young to middle-aged populations [38,39,40]. However, the optimal RC level for preventing CVD in young to middle-aged populations has not yet been reported. The present study suggested increased risks of CVD, CHD, and fatal CVD for participants 35 to 65 years of age and an RC level > 0.83 mmol/L. This may contribute to risk stratification for dyslipidemia and constructing further treatment strategies to prevent CVD in young to middle-aged populations.

DM confers at least a two to threefold excess risk for ASCVD [41]. Data have revealed that serum RC concentrations are elevated in patients with DM or pre-DM and can increase the risk for CHD and future coronary outcomes [42,43,44]. This suggests RC as a possible treatment target for patients with impaired glucose metabolism [17]. A baseline RC level ≥ 30 mg/dL (0.78 mmol/L) identifies a high risk of major adverse cardiovascular events for individuals with type 2 DM (T2DM) or more than three CVD risk factors [45]. Additionally, the risk is independent of whether LDL-C levels are on target at ≤100 mg/dL (2.59 mmol/L) or not. For patients with T2DM who underwent previous percutaneous coronary stents, a baseline RLP-C level at 0.505 mmol/L was identified as the optimal cutoff point to predict in-stent restenosis [46]. The lipid level for participants with DM should be lower than those without DM for preventing ASCVD. Several guidelines proposed a stratification of cardiovascular risk among people with DM and recommend an intensive therapy for dyslipidemia in DM [47,48]. However, the target for regulating RC concentration is still not assured among Chinese populations with DM. In this study, the incidence of combined CVD and CHD significantly increased for general Chinese participants with a T2DM and serum level of RC > 0.95 mmol/L (Figure 4). It follows that RC is another intervening target to reduce CVD except LDL-C.

The mechanism behind this cardiovascular benefit for RC is not fully understood. There is a complex link between RC, inflammation, and CVD events [49]. Balling evidence revealed that one-third of TC in plasma was present in RC by using direct measurements [50]. High levels of RC can cause a variety of proatherogenic effects, including monocyte activation, upregulation of proinflammatory cytokines, and increased prothrombotic factors production [26,51]. Remnant lipoproteins can enter and get trapped in the intima of the arterial wall, promoting endothelial dysfunction and inflammation through increased secretion of various cytokines and adhesion molecules [52]. Without oxidative modification by macrophages [53], RC is accumulated in the arterial wall and may play a causal role in the development of atherosclerosis and ultimately ASCVD [26,27]. RC also accelerates the onset of endothelial progenitor cells senescence via increased oxidative stress and induces endothelial dysfunction by inhibiting nitric oxide production [54]. Autopsy case analysis showed coronary atherosclerosis in the young (<40 years), commonly exhibiting eccentric plaques with associated inflammation [55]. The atherosclerotic plaques of younger patients are rich in foam cells, which reduce plaque stability and induce early-onset ACS [56]. Mazzone reviewed the important roles of diabetic dyslipidemia (beyond the LDL cholesterol level) and inflammation for accelerating vascular injury and increasing the rates of CVD in T2DM patients [57].

The present study demonstrated that high levels of RC were significant predictors of increased risk of CVD events, especially for the 35- to 65-years-old population and DM patients. This suggests that assessing RC levels in these populations might be likely to have clinical utility in terms of CVD risk stratification and future intervention. Further investigations into RC lowering interventions to reduce residual ASCVD risk are necessary. This study had several limitations. Firstly, RC was calculated with the Friedewald formula but not measured directly. A simple and widely available assay is needed to be developed to measure the cholesterol content of RC. Secondly, the effect of different constructions of diet on RC concentration is lacking. Finally, this was a study among a general Chinese population. Studies of RC in other countries with different diet cultures need further investigation.

## 5. Conclusions

In conclusion, in this large-scale population-based study, we found a high level of RC significantly related to a worse prognosis. For the first time, we revealed that the risk of CVD significantly increased after the median values of RC for young to middle-aged populations (0.83 mmol/L) and the DM subgroup (0.95 mmol/L). RC as a causative risk factor for CVD events deserves further attention and may constitute a prominent target for interventions to reduce vascular risk after LDL cholesterol lowering, especially for young to middle-aged populations or participants with diabetes.

## Figures and Tables

**Figure 1 jcm-10-03388-f001:**
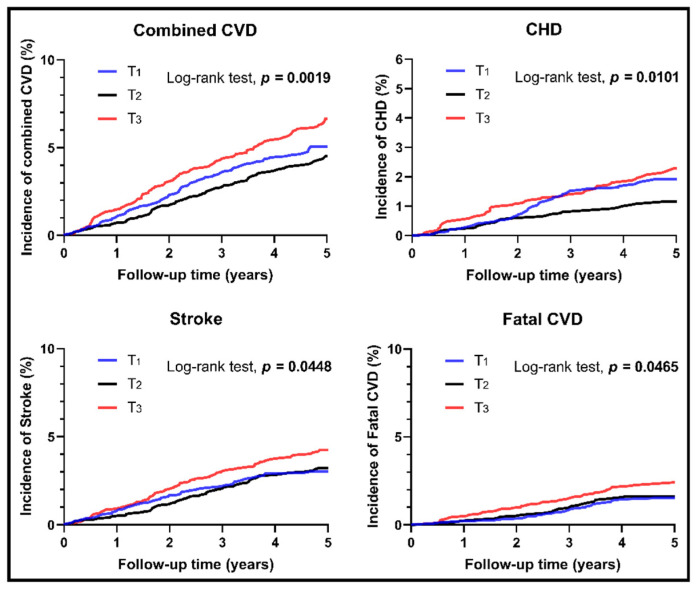
Unadjusted Kaplan–Meier curves for incident cardiovascular events stratified by different levels of remnant cholesterol. The whole cohort was divided into three groups due to different levels of remnant cholesterol. Low level: Tertile I; Medium level: Tertile II; High level: Tertile III. CHD: coronary heart disease; CVD: cardiovascular disease.

**Figure 2 jcm-10-03388-f002:**
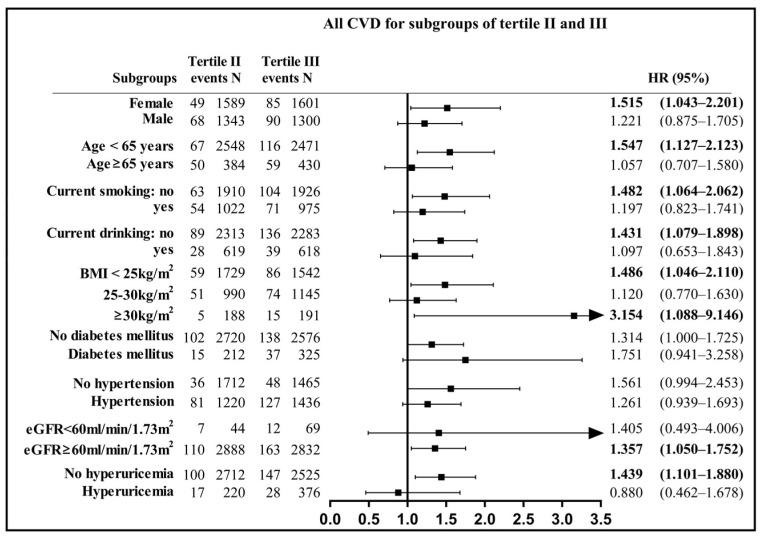
Remnant cholesterol (Tertile II and Tertile III) in relation to combined cardiovascular disease (CVD) for different subgroups. Model adjusted for age, sex, ethnicity, body mass index, smoking status, drinking status, hypertension, diabetes mellitus, estimated glomerular filtration rate, hyperuricemia, and continuous triglyceride. BMI: body mass index; CI: confidence interval; CVD: cardiovascular disease; eGFR: estimated glomerular filtration rate; HR: hazard ratio; RC: remnant cholesterol.

**Figure 3 jcm-10-03388-f003:**
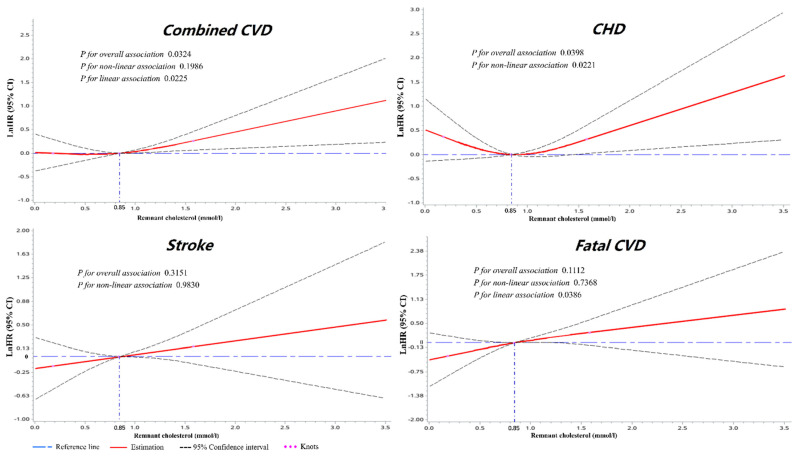
The dose–response associations between RC and cardiovascular outcomes for the whole cohort. Restricted cubic splines displaying the lnHRs of cardiovascular events with 95% confidence intervals according to the serum level of remnant cholesterol. Reference set to median (0.84 mmol/L). Knots located at 0.18, 0.84, and 1.58 mmol/L (at the 5th, 50th, and 95th percentiles). Adjusted for age, sex, ethnicity, body mass index, smoking status, drinking status, hypertension, diabetes mellitus, estimated glomerular filtration rate, hyperuricemia, and continuous triglyceride. CHD: coronary heart disease; CVD: cardiovascular disease; lnHRs: natural log of hazard ratios; RC: remnant cholesterol.

**Figure 4 jcm-10-03388-f004:**
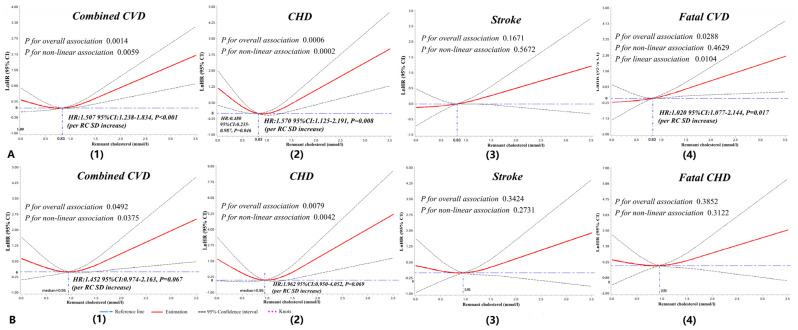
Dose-response associations between RC and cardiovascular outcomes for subgroups aged < 65 years and DM. Restricted cubic splines displaying the lnHRs of cardiovascular events among participants aged < 65 years (**A**(1–4)) and diabetes (**B**(1–4)) with 95% confidence intervals according to the serum level of remnant cholesterol. Reference set to medians (0.83 mmol/L for subgroups aged < 65 years, 0.95 mmol/L for the diabetes mellitus subgroup). Knots located at the 5th, 50th, and 95th percentiles of remnant cholesterol (subgroup aged < 65 years: 0.17, 0.83, and 1.57 mmol/L; DM: 0.25, 0.95, and 1.76 mmol/L). Adjusted for age, sex, ethnicity, body mass index, smoking status, drinking status, hypertension, DM, estimated glomerular filtration rate, hyperuricemia, and continuous triglyceride. CHD, coronary heart disease; CVD, cardiovascular disease; DM, diabetes mellitus; HR, hazard ratio; lnHRs, natural log of hazard ratios; RC, remnant cholesterol; SD, standard deviation.

**Table 1 jcm-10-03388-t001:** Baseline characteristics by tertiles of remnant cholesterol.

	Total *n* = 8782	Remnant Cholesterol			*p*-Value	*p* _1_	*p* _2_	*p* _3_
		Tertile I (*n* = 2949)	Tertile II (*n* = 2932)	Tertile III (*n* = 2901)				
Age (year)	53.2 ± 10.4	52.2 ± 10.6	52.7 ± 10.4	54.7 ± 10.0	<0.001	0.201	<0.001	<0.001
Male (%)	4075 (46.4)	1432 (48.6)	1343 (45.8)	1300 (44.8)	0.012			
Ethnicity of Han (%)	8277 (94.2)	2664 (90.3)	2820 (96.2)	2793 (96.3)	<0.001			
Current smoking (%)	3118 (35.5)	1121 (38.0)	1022 (34.9)	975 (33.6)	0.001			
Current drinking alcohol (%)	1992 (22.7)	755 (25.6)	619 (21.1)	618 (21.3)	<0.001			
Physical activity								
Low	2964 (33.8)	819 (27.8)	1033 (35.2)	1112 (38.3)	<0.001			
Medium	1652 (18.8)	566 (19.2)	581 (19.8)	505 (17.4)				
High	4091 (46.6)	1534 (52.0)	1290 (44.0)	1267 (44.0)				
HTN (%)	4154 (47.3)	1141 (38.7)	965 (32.9)	911 (31.4)	<0.001			
DM (%)	738 (8.4)	201 (6.8)	212 (7.2)	325 (11.2)	<0.001			
BMI (kg/m^2^)	24.7 ± 3.6	24.7 ± 3.8	24.4 ± 3.6	24.8 ± 3.5	<0.001	0.003	0.866	<0.001
SBP (mmHg)	140.5 ± 22.7	144.0 ± 24.3	137.1 ± 21.6	140.3 ± 21.5	<0.001	<0.001	<0.001	<0.001
DBP (mmHg)	81.6 ± 11.5	81.1 ± 11.8	81.1 ± 11.5	82.6 ± 11.3	<0.001	1.0	<0.001	<0.001
FBG (mmol/L)	5.8 ± 1.5	5.6 ± 1.4	5.8 ± 1.7	5.8 ± 1.3	<0.001	0.043	<0.001	<0.001
TC (mmol/L)	5.2 ± 1.0	4.7 ± 0.9	5.0 ± 0.8	5.8 ± 1.0	<0.001	<0.001	<0.001	<0.001
TG (mmol/L)	1.4 ± 0.8	1.1 ± 0.5	1.3 ± 0.7	1.8 ± 0.8	<0.001	<0.001	<0.001	<0.001
HDL-C (mmol/L)	1.4 ± 0.3	1.6 ± 0.4	1.3 ± 0.3	1.3 ± 0.3	<0.001	<0.001	<0.001	0.626
LDL-C (mmol/L)	2.9 ± 0.8	2.8 ± 0.8	2.8 ± 0.7	3.1 ± 0.8	<0.001	0.429	<0.001	<0.001
Estimated eGFR (mL/min/1.73 m^2^)	94.1 ± 15.1	101.6 ± 13.2	92.3 ± 14.3	88.2 ± 14.4	<0.001	<0.001	<0.001	<0.001
Serum uric acid (μmol/L)	285.6 ± 81.5	260.7 ± 73.8	284.8 ± 79.5	290.4 ± 78.5	<0.001	<0.001	<0.001	<0.001
RC (mmol/L)	0.83 ± 0.44	0.36 ± 0.16	0.84 ± 0.10	1.32 ± 0.30	<0.001	<0.001	<0.001	<0.001

BMI: body mass index; DBP: diastolic blood pressure; DM: diabetes mellitus; FPG: fasting plasma glucose; GFR: glomerular filtration rate; HDL-C: high-density lipoprotein cholesterol; HTN: hypertension; LDL-C: low-density lipoprotein cholesterol; RC: remnant cholesterol; SBP: systolic blood pressure; SD: standard deviation; TC: total cholesterol; TG: triglyceride. Data are expressed as mean ± SD or as *n* (%). *p*-Value: statistical significance among three groups; *p*_1_, *p*_2_, and *p*_3_: respective statistical significance with Bonferroni correction for measurement data between Tertile I and Tertile II, Tertile I and Tertile III, and between Tertile II and Tertile III.

**Table 2 jcm-10-03388-t002:** Multivariate-adjusted hazard ratios and 95% confidence intervals for cardiovascular outcomes associated with baseline remnant cholesterol.

	RC Q_1_	RC Q_2_	RC Q_3_	RC Continuous (1-SD ^†^)
Combined CVD				
*n*/*N*	139/2191	117/2932	175/2901	431/8782
Model 1	1.23 (0.96–1.57)	1.00 (ref)	1.49 (1.18–1.89) **	1.33 (1.09–1.63) **
Model 2	1.22 (0.95–1.57)	1.00 (ref)	1.49 (1.18–1.89) **	1.32 (1.08–1.62) **
Model 3	1.16 (0.90–1.50)	1.00 (ref)	1.37 (1.07–1.74) *	1.28 (1.02–1.62) *
CHD				
*n*/*N*	55/2191	34/2932	61/2901	150/8782
Model 1	1.77 (1.15–2.74) *	1.00 (ref)	1.71 (1.12–2.61) *	1.17 (0.82–1.67)
Model 2	1.74 (1.12–2.70) *	1.00 (ref)	1.71 (1.12–2.61) *	1.16 (0.81–1.66)
Model 3	1.68 (1.08–2.62) *	1.00 (ref)	1.63 (1.06–2.53) *	1.15 (0.76–1.74)
Stroke				
*n*/*N*	88/2191	87/2932	118/2901	293/8782
Model 1	1.04 (0.77–1.40)	1.00 (ref)	1.30 (0.98–1.72)	1.31 (1.03–1.67) *
Model 2	1.04 (0.77–1.41)	1.00 (ref)	1.31 (0.98–1.73)	1.30 (1.02–1.67) *
Model 3	0.99 (0.73–1.35)	1.00 (ref)	1.19 (0.89–1.59)	1.25 (0.94–1.66)
Fatal CVD				
*n*/*N*	41/2191	44/2932	63/2901	148/8782
Model 1	0.97 (0.64–1.49)	1.00 (ref)	1.394 (0.95–2.05)	1.44 (1.05–1.97) *
Model 2	0.95 (0.62–1.46)	1.00 (ref)	1.421 (0.97–2.09)	1.47 (1.07–2.01) *
Model 3	0.90 (0.58–1.39)	1.00 (ref)	1.37 (0.92–2.05)	1.51 (1.05–2.17) *

CHD: coronary heart disease; CVD: cardiovascular disease; RC: remnant cholesterol. Model 1: adjusted for age (<65 years vs. ≥65 years), sex, and ethnicity (Han or not). Model 2: adjusted for factors in Model 1 and smoking status, drinking status, and body mass index (normal, overweight, obesity). Model 3: adjusted for factors in Model 2 and estimated glomerular filtration rate (<60 mL/min/1.73 m^2^ vs. ≥60 mL/min/1.73 m^2^), diabetes mellitus (yes or no), hypertension (yes or no), triglyceride (continuous), and hyperuricemia (yes or no). ^†^ HR for continuous 1-SD increment. * *p* < 0.05; ** *p* < 0.01.

## Data Availability

All data generated or analyzed during this study are included in this published article (and its Appendix A). The datasets generated during and/or analyzed during the current study are not publicly available due to the lack of a specific patients’ consent but are made available by the corresponding author based on a reasonable request.

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
