# Peer review of "The Effects of Calculated Remnant-Like Particle Cholesterol on Incident Cardiovascular Disease: Insights from a General Chinese Population"

_jcm, 2021, doi:10.3390/jcm10153388_

Round 1

Reviewer 1 Report

This manuscript is a prospective cohort study investigating remnant cholesterol in the Chinese population. It is a well performed and well written study with a good number of participants. 

There are only a couple minor clarifying things that I would like addressed:

  1. Why is the follow-up only 4.66 years versus 5 years? Was there a reason it was cut short? Or was this the plan from the beginning?
  2. What was the timing of the blood samples? Were patients cholesterol tested more than once throughout the study period? At what intervals? 
  3. How did you determine the numbers for the Tiers? Are you suggesting that these numbers should be how we determine the cut off point to treat (as you mention in your abstract)?
  4. What is the accuracy of the Friedewald formula?

Author Response

1st Reviewer’s comments:
Comment 1: English language and style are fine/minor spell check required.
Response 1: Thank you for your comments! We have revised the whole manuscript. Thank you again. Some of the abbreviations were revised again (Page 1, Line 19, 28; Page 2, Line 67-68, 95-97; Page 3, Line 131, 141, 146; Page 4, Line 212, 230; Page 5, Line 268; Page 7, Line 353; Page 9, Line 516; Page 10, Line 566-568). The discussion part was carefully checked to be more reasonable (Page 9, Line 535-537; Page 10, Line 551-555, Line 565-572, Line 585-590). The third paragraph in Page 10 was revised, and the references order was changed accordingly in the reference part. Meanwhile, some grammar of sentence was checked again. Thank you.
Comment 2:
Why is the follow-up only 4.66 years versus 5 years? Was there a reason it was cut short? Or was this the plan from the beginning?

Response 2: Thank you for your good comments. We conducted the follow-up visits irregularly for this cohort of 11,956 subjects. The visit from 2017 to 2018 was performed before another intervention study begun (ClinicalTrials.gov, NCT03527719). As a result, the median follow-up time was only 4.66 years. Accordingly, a sentence of “A series of irregular follow-up visits were conducted” was added in Line 150 of Page 3 in the marked manuscript. Recently, we are planning to conduct a new follow-up visit for this cohort. The following results will be considered to submit to JOURNAL OF CLINICAL MEDICINE. Thank you for your comments.
Comment 3
: What was the timing of the blood samples? Were patients’ cholesterol tested more than once throughout the study period? At what intervals? 

Response 3: Thank you for your good comment! All blood samples were collected in the morning after at least 12 hours’ fasting for all subjects. In order to describe it clearly, the sentence in Line 123 of Page 3 in the marked manuscript was changed to “Fasting blood samples were collected in the morning after at least 12 h fasting for all subjects”. In the present study, blood cholesterol tests were performed irregularly. All subjects had their fasting lipid profile at the first (in 2013) and the second follow-up (in 2015). The third follow-up visit was to collect cardiovascular events, and no blood tests and research questionnaires were performed. During follow-up in further, regular follow-up and blood tests with three years’ interval will be performed to investigate the trajectory of some cardiovascular diseases. Thank you for your helpful comments.

Comment 4: How did you determine the numbers for the Tiers? Are you suggesting that these numbers should be how we determine the cut off point to treat (as you mention in your abstract)?

Response 4: Thank you for your comments! According to the different level of remnant cholesterol (RC) (low, medium, and high), all subjects were divided into three groups with the 33.3% and 66.6% percentiles of RC. All baseline data were compared between these three groups. Accordingly, the sentence in Line 156-158 of Page 3 in marked manuscript was revised to “According to the 33.3% and 66.6% percentiles of RC level, all subjects were divided into three groups (tertile I, tertile II, and tertile III) and the baseline characteristics were assessed across different groups”. Based on close relationships between cardiovascular events and RC groups (Figure 1, Figure 2, and Table 2), the dose-response analyses between RC (continuous variable) and outcomes were performed to determine the cutoff points to treat RC in different population (Figure 3 and Figure 4). Restricted cubic spline regression was applied to do the non-linear association between continuous RC and outcomes. The knots were placed at the 5th, 50th, and 95th percentiles of RC. In Figure 3 and Figure 4, point marked on abscissa was the median of RC in each subgroup. Significant associations between RC level and outcomes were found for some subgroups around curve break points (Figure 3: combined CVD, CHD; Figure 4: combined CVD, CHD, and fatal CVD for participants with age < 65 years; Figure 4: combined CVD and CHD for participants with diabetes). As a result, the curve break points or the median of RC in subgroup were considered as the cutoff points to reduce cardiovascular outcomes. This method has been shown by other study.[1] Thank you very much for your insightful comments.

Comment 5: What is the accuracy of the Friedewald formula?

Response 5: Thank you for your comment! There is no study relating to the accuracy of the Friedewald formula until now. Several ways of measuring the concentration of RC have been reported, including ultracentrifugation, direct RC assays, or nuclear magnetic resonance (NMR) spectroscopy.[2] But these methods are labor-intensive and relatively expensive. Neither of which is broadly applied for routine clinical use. At present, there is no fully automated, commercial assay available that measures RC accurately, corresponding to calculated RC. Meanwhile, calculated RC has also been reported close associations with atherosclerosis or cardiovascular outcomes by many studies. [3-7] It could be speculated that calculated RC and directly measured RC provide similar credibility of all-cause mortality risk, and maybe also risk of cardiovascular disease. And the calculated RC is most easily accessible and therefore could be preferred in the clinic. So, in the present study, calculated RC was applied to investigate the relationships with cardiovascular outcomes. Your opinion will contribute to the future research.

Based on the instructions provided in your letter, we uploaded the file of the revised manuscript with all the changes highlighted by using the track changes mode in MS Word named “Main Document - marked copy” and a clean copy named “Main Document”. Appended to this letter is our point-by-point response to the comments raised by the reviewers.

Thank you again for your positive and constructive comments and suggestions on our manuscript. 

Reviewer 2 Report

The authors investigate the relationships between remnant cholesterol and combined cardiovascular diseases in a cohort from China with age interval 35-65. The study started with more than 11000 subjects and after the follow ups reduced to 8782 subjects. Number of the participants are quite high and sufficient for the study. The paper is written well and results are presented clearly. The methods are explained well. The conclusions are supported with the results. However, the submitted manuscript requires revisions.

1.There are three groups in the statistical analysis where the authors correctly used ANOVA for the samples distributed normally and Kruskal-Wallis for the samples distributed in non-normal way. Did they perform post-hoc tests such as student t-test or Mann-Whitney U test for the data distributed in normal and non-normal ways to assess the differences between each group after ANOVA and Kruskal-Wallis tests?

2. The authors present hazard ratios in forest plots. It can be good to show subtotal values for each subgroup in the forests plots. Especially for the subgroups such as hyperuricemia where different events are favored.

Minor comment: The figure legends are not included in the text in the main file whereas the figure legends are available only for the figures in the supplementary materials. They should be added.

Author Response

2nd Reviewer’s comments:
Comment 1: English language and style are fine/minor spell check required.
Response 1: Thank you for your comments! We have revised the whole manuscript. Thank you again. Some of the abbreviations were revised again (Page 1, Line 19, 28; Page 2, Line 67-68, 95-97; Page 3, Line 131, 141, 146; Page 4, Line 212, 230; Page 5, Line 268; Page 7, Line 353; Page 9, Line 515; Page 10, Line 565-567). The discussion part was carefully checked to be more reasonable (Page 9, Line 534-536; Page 10, Line 549-553, Line 563-570, Line 583-588). The third paragraph in Page 10 was revised, and the references order was changed accordingly in the reference part. Meanwhile, some grammar of sentence was checked again. Thank you.  

Comment 2: The results can be improved.

Response 2: Thank you for your carefully work! We checked the “Results” part again. It's true that there were some confusions in this part. The paragraph from line 258 to line 265 in page 5 was revised to clarify the inclination of baseline data across tertiles of remnant cholesterol. In page 5, several negative results were deleted. Thank you again for your good advice!

Comment 3: There are three groups in the statistical analysis where the authors correctly used ANOVA for the samples distributed normally and Kruskal-Wallis for the samples distributed in non-normal way. Did they perform post-hoc tests such as student t-test or Mann-Whitney U test for the data distributed in normal and non-normal ways to assess the differences between each group after ANOVA and Kruskal-Wallis tests?

Response 3: Thank you for your carefully work! We performed post-hoc tests for ANOVA or Kruskal-Wallis, respectively for normally distributed data or non-normal data. But P-values for post-hoc analyses were not listed in previous manuscript. According to your opinion, all measurement data was analyzed using post-hoc analysis with Bonferroni correction again, and P values were added in Table. 1. Accordingly, the results of “baseline characteristics” were revised to add the results of post-hoc analysis (Line262-265 in page 5). Thank you again for your good comments which helps to improve the paper.

Comment 4: The authors present hazard ratios in forest plots. It can be good to show subtotal values for each subgroup in the forests plots. Especially for the subgroups such as hyperuricemia where different events are favored.

Response 4: Thank you for your helpful assessment. We have evaluated subtotal values for each subgroup in the forests plots and them in Figure 2 and Supplementary Figures (2-4). Thank you.

Comment 5: The figure legends should be added.

Response 5: Thank you for your carefully work! All figures’ legends were lost during submission. Now we added all figures’ legends again in the main file. Thank you for your patience.

Based on the instructions provided in your letter, we uploaded the file of the revised manuscript with all the changes highlighted by using the track changes mode in MS Word named “Main Document - marked copy” and a clean copy named “Main Document”. Appended to this letter is our point-by-point response to the comments raised by the reviewers.

Thank you again for your positive and constructive comments and suggestions on our manuscript. 

Reviewer 3 Report

Comment 1. The concentration of RC was calculated, but not directly measured. So, it may be better to change RC to Calculated RC as shown in Cardiovasc Diabetol. 2020 Jul 6;19(1):104.

Comment 2. Which is correct; mean follow-up of 4.66 years (abstract) or median (Results 3.2.)? Median may be better.

Comment 3. In all subgroup analyses, authors should add all P values for interaction (e.g. female vs male).

Comment 4. Figure 3 and 4 are hard to see.

Comment 5. Authors should add figure legends.  

Comment 6. This reviewer cannot understand why authors selected only age and DM in a non-linear association analysis (Fig 4). How about subgroups, including hypertension, BMI, drinking, and hyperuricemia?

Author Response

3rd Reviewer’s comments:
Comment 1: The concentration of RC was calculated, but not directly measured. So, it may be better to change RC to Calculated RC as shown in Cardiovasc Diabetol. 2020 Jul 6;19(1):104.
Response 1: Thank you for your valuable comments! We have changed “RC” to “calculated RC” for the whole manuscript as shown in the reference [Cardiovasc Diabetol. 2020 Jul 6;19(1):104], including “RC” in Title, Line 199 in Page 4, Line 516 in Page 9, and Line 595 in Page 10. Thank you again.

Comment 2: Which is correct; mean follow-up of 4.66 years (abstract) or median (Results 3.2.)? Median may be better.

Response 2: Thank you for your carefully work! As you said, the “median” is better for the definition of follow-up years or survival time. The median follow-up years were 4.66 years. So “mean” in line 20 of abstract was corrected to “median”. Thank you for your advice.

Comment 3: In all subgroup analyses, authors should add all P values for interaction (e.g. female vs male).

Response 3: Thank you for your valuable comments! The interaction P values should be calculated for all subgroups. We reviewed all basic characteristics, and calculated the interaction p values for subgroup factors and remnant cholesterol on the association with all outcomes, with adding “a*b” into covariates of cox regression. As a result, no significant interactions were found for all subgroups and RC on outcomes. Because of no significant interaction was resulted, interaction P values were only described in the results of “Stratification analyses” (Lines 358-359 and Line 364-366 in Page 7), and not shown in Figure 2. Thank you for your help again.

Comment 4: Figure 3 and 4 are hard to see.

Response 4: Thank you for your comments! Figure 3 and 4 are large with high pixel, due to the author guides of JOURNAL OF CLINICAL MEDICINE. We are sorry for that they are difficult to be opened. In the revised manuscript, all figures may be easy to be opened due to the editor’s help on pictures typesetting. Meanwhile, figures 3 and 4 and their corresponding legends were resubmitted with PDF format to be reviewed easily. We are sorry for the inconvenience to you. Thank you.

Comment 5: The figure legends should be added.

Response 5: Thank you for your carefully work! All figures’ legends were lost during submission. Now we added all figures’ legends again in the main file. Thank you for your patience.

Comment 6: This reviewer cannot understand why authors selected only age and DM in a non-linear association analysis (Fig 4). How about subgroups, including hypertension, BMI, drinking, and hyperuricemia?

Response 6: Thank you for your good comments! In fact, we analyzed dose-response associations between RC and outcomes for all subgroups (including gender, age, hypertension, diabetes, BMI, drinking, smoking, hyperuricemia, and renal dysfunction). Both significant overall and non-linear association results were found for subgroups age and diabetes. Although significant overall, non-linear, or linear associations were also found for some other subgroups (female, non-smokers, non-drinkers, or non-hyperuricemia), only data of subgroups DM and age were listed in the paper due to the limit of whole manuscript and for the sake of discussion. According to the reviewers’ valuable comments, other significant associations can be listed in Supplementary Figure 5 (Line 485-486 in Page 8). All these Figures (Affiliated Figures) were also shown at the end of this document. Thank you for your valuable comments again.

Based on the instructions provided in your letter, we uploaded the file of the revised manuscript with all the changes highlighted by using the track changes mode in MS Word named “Main Document - marked copy” and a clean copy named “Main Document”. Appended to this letter is our point-by-point response to the comments raised by the reviewers.

Thank you again for your positive and constructive comments and suggestions on our manuscript.

Round 2

Reviewer 3 Report

Thank you for your prompt reply.

Authors addressed my comments adequately.